# Association between dietary inflammation index and herpes simplex virus I and II: A cross-sectional study

Jing Luo[1,2��], En-Hui Liu[1,4��], Hao-Kai Chen[1,2��], Xiang-Ping He[1,5], Tong Chen[1,2], Yu-Qi Hu[1,2], Xu-Guang Guo[1,2,3] *

**1** Department of Clinical Laboratory Medicine, Guangdong Provincial Key Laboratory of Major Obstetric Diseases, Guangdong Provincial Clinical Research Center for Obstetrics and Gynecology, The Third Affiliated Hospital, Guangzhou Medical University, Guangzhou, China, **2** The Third School of Clinical Medicine, Guangzhou Medical University, Guangzhou, China, **3** Guangzhou Key Laboratory for Clinical Rapid Diagnosis and Early Warning of Infectious Diseases, King Med School of Laboratory Medicine, Guangzhou Medical University, Guangzhou, China, **4** School of Pediatrics, Guangzhou Medical University, Guangzhou, China, **5** The Clinical School of Integrated Traditional Chinese and Western Medicine, Guangzhou Medical University, Guangzhou, China

☯ These authors contributed equally to this work.
* gysygxg@gmail.com

## Abstract

### Introduction

We aimed to fill the research gap between DII and herpes simplex virus infection among adults in the US by analyzing the association between dietary inflammatory index and herpes simplex virus and to provide new ideas for herpes simplex virus prevention and treatment.

### Method

We used data from 8636 participants in NHANES 2007–2016, which were statistically analyzed by participant baseline study, one-way analysis of variance, multiple regression equations, smoothed curve fitting, and stratified analysis.

### Result

In the fully adjusted model, the DII high concentration group was positively associated with the prevalence of herpes simplex (1.15 (0.89, 1.48), p = 0.0027), and the results of the stratified analyses indicated that the positive association between DII and herpes simplex virus type II was stable in the population.

### Conclusion

This study demonstrates a positive association between DII and herpes simplex virus II in US adults, suggesting that a proinflammatory diet may be an independent risk factor for herpes simplex virus II.

**Funding:** The author(s) received no specific funding for this work.

**Competing interests:** The authors have declared that no competing interests exist.

## Introduction

Most people have been infected by HSV and carry reactivatable latent viruses (Whitley RJ, Roizman B. Herpes simplex virus infections. The Lancet. 2001 May 12;357(9267):1513–8. DOI: 10.1016/S0140-6736 (00)04638-9. PMID: 11377626.); it is estimated that one-third of patients have more than six recurrences per year, one-third have two recurrences per year, and the remaining one-third have rare recurrences. (Mertz GJ, Benedetti J, Ashley R, Selke SA, Corey L. Risk factors for sexual transmission of genital herpes. Ann Intern Med. DOI: 10.7326/0003-4819-116-3-197. PMID: 1309413.) Some studies have confirmed an association between an inflammatory diet and acute and chronic disease risk (Morshedzadeh N, Rahimlou M, Shahrokh S, Karimi S, Mirmiran P, Zali MR. Effects of flaxseed supplementation on meta-bolic syndrome markers, insulin resistance, and inflammation in patients with ulcerative coli-tis: an open-label, randomized, controlled trial. doi. 10.1002/ptr.7081. Epub 2021 Apr 15. PMID: 33856729.).

A study suggests a relationship between the ketogenic diet and herpes simplex virus infec-tion [1]. The Dietary Inflammation Index (DII) assesses the inflammatory potential of a diet and can categorize an individual's diet from maximal anti-inflammatory to maximal proin-flammatory effects [2]. The DII is based on a large body of literature and research exploring the effects of diet on inflammation in the areas of cell cultures, animal experiments, and epide-miological studies, and the DII comprehensively takes into account the entire dietary picture rather than just individual nutrients or foods [3]. Based on an extensive literature search, the diet of a population causes changes in the levels of a number of inflammatory markers in the body, ultimately focusing on the following inflammatory markers: IL-1β, IL-4, IL-6, IL-10, TNFα, and CRP [4]. High DII scores indicate a stronger pro-inflammatory diet, and low DII scores indicate a stronger anti-inflammatory diet. The food parameter was assigned a "+1" [5] if the effect of the food parameter was pro-inflammatory (significantly increased IL-1β, IL-6, TNF-α, or CRP, or lowered IL-4 or IL-10); "+1" was assigned if the effect was anti-inflamma-tory (significantly lowered IL-1β, IL-6, TNF, or CRP, or increased IL; for an anti-inflammatory effect -1" (significant decrease in IL-1β, IL-6, TNF or CRP, or increase in IL-4 or IL-10); and "0" if it did not cause a change in inflammatory markers. The DII is better able to guide and assess individual dietary goals to reduce levels of inflammation and the harms of some chronic diseases [6].

There is a highly prevalent virus in humans called Herpes simplex virus (HE) [7]. HE can be categorized into two types, HE-1 and HE-2, with global prevalence rates of 67% and 13%, respectively [8]. HE can be transmitted through close contact, leading to lifelong infection, and latent infections are also present in neuronal cells and are capable of causing a wide range of diseases, including oral and labial herpes, genital herpes, hemorrhagic keratitis, meningitis and encephalitis [9]. HE-I primarily causes oral and labial herpes, while HE-II causes genital herpes almost exclusively through sexual transmission, and some studies have shown that both types of HE may place infected individuals at a higher risk for HIV infection [10]. HE infection is capable of inducing innate and adaptive immunity, causing cells to produce inflammatory fac-tors, such as IL-6, which results in a certain amount of inflammatory response in the organism [11].

There is a lack of research on the relationship between DII and HSV infection in U.S. adults. Although the relationship between DII and multiple health outcomes has been exten-sively studied, research linking it to the risk of HSV infection is extremely limited. Therefore, this study fills this gap in the field and provides a potential basis for future nutritional interven-tion strategies for HSV infection. NHANES is a nationally representative database of rich die-tary and health data that is ideally suited for this study to help reveal the potential relationship

between DII and HSV infection in different population subgroups. We constructed a cross-sectional study including 8,636 participants based on the National Health and Nutrition Examination Survey (NHANES) from 2007 to 2016 with the aim of exploring the association between DII and HE.

## Methods

### Data sources

A five-phase study from the National Health and Nutrition Examination Survey (NHANES) was used in this study. The five phases used were 2007–2008, 2009–2010, 2011–2012, 2013–2014, and 2015–2016 including 8,636 participants. The health and nutritional status of a diverse population in the U.S. are assessed as part of the NHANES cross-sectional study. It collects a large amount of data on diet, nutritional status, and chronic diseases, among other comprehensive data. Among them, questions related to demographics, socioeconomics, education, age, diet and health were obtained in the form of questionnaires, while most of the physical examinations, including laboratory results such as the presence or absence of HE infection and VC levels in the present study, were obtained at mobile examination centers (MECs). A written informed consent form was completed by all study participants, and the NCHS Research Ethics Review Board (https://wwwn.cdc.gov/nchs/nhanes/default.aspx) approved this study.

A total of 50,588 data points were included in our study. We excluded 14,657 data points remaining after 35,931 data points that were unsure of whether they were infected with HE-1 and HE-2, 13,756 data points remaining after an additional 901 data points that did not have DII, and 8,636 data points remaining after 5,120 data points for which the covariates were unknown. Detailed troubleshooting flowchart in Fig 1.

### Calculation of the dietary inflammation index

The DII is a scoring system that assesses the potential inflammatory levels of dietary components. The DII calculates the effects of 45 nutrients on the inflammatory response, and the methodology for calculating the DII was developed by Shivappa through a comprehensive evaluation of peer-reviewed articles. The DII is based on the addition of scores for each component from diets consumed over a 24-hour period, including scores from pro- and anti-inflammatory diets. The DII is calculated by subtracting the daily intake of each nutrient from the global daily intake of each nutrient. The Z score is calculated by subtracting the global mean intake from the standard deviation and dividing by it, where the standard deviation represents the individual's exposure relative to the "standard global mean. Each Z score was then converted to a percentile score. The central percentile score was then calculated by doubling each percentile score and subtracting one from it to achieve a symmetrical distribution, and the central percentile score for each food parameter was multiplied by the "overall food parameter-specific inflammatory effect score" to produce the following results. To determine the food parameter-specific DII score, the central percentile score for each food parameter was multiplied by the "overall food parameter-specific inflammatory effect score" for that food parameter. The DII scores for each participant were summed to obtain a separate overall DII score. In this research, DII scores were calculated for 28 available food parameters: carbohydrate, total fat, protein, alcohol, cholesterol, fiber, MUFA, saturated fat, PUFA, n-6 fatty acids, niacin, n-3 fatty acids, vitamin A, vitamin B2, vitamin B6, vitamin B1, vitamin B12, vitamin D, vitamin E, vitamin C, Fe, Mg, selenium, zinc, folate, caffeine, beta-carotene, and energy. We categorized the DII into three groups: low (Low: -2 to 0), middle (Middle: 0 to 2), and high (High: 2 to 4).

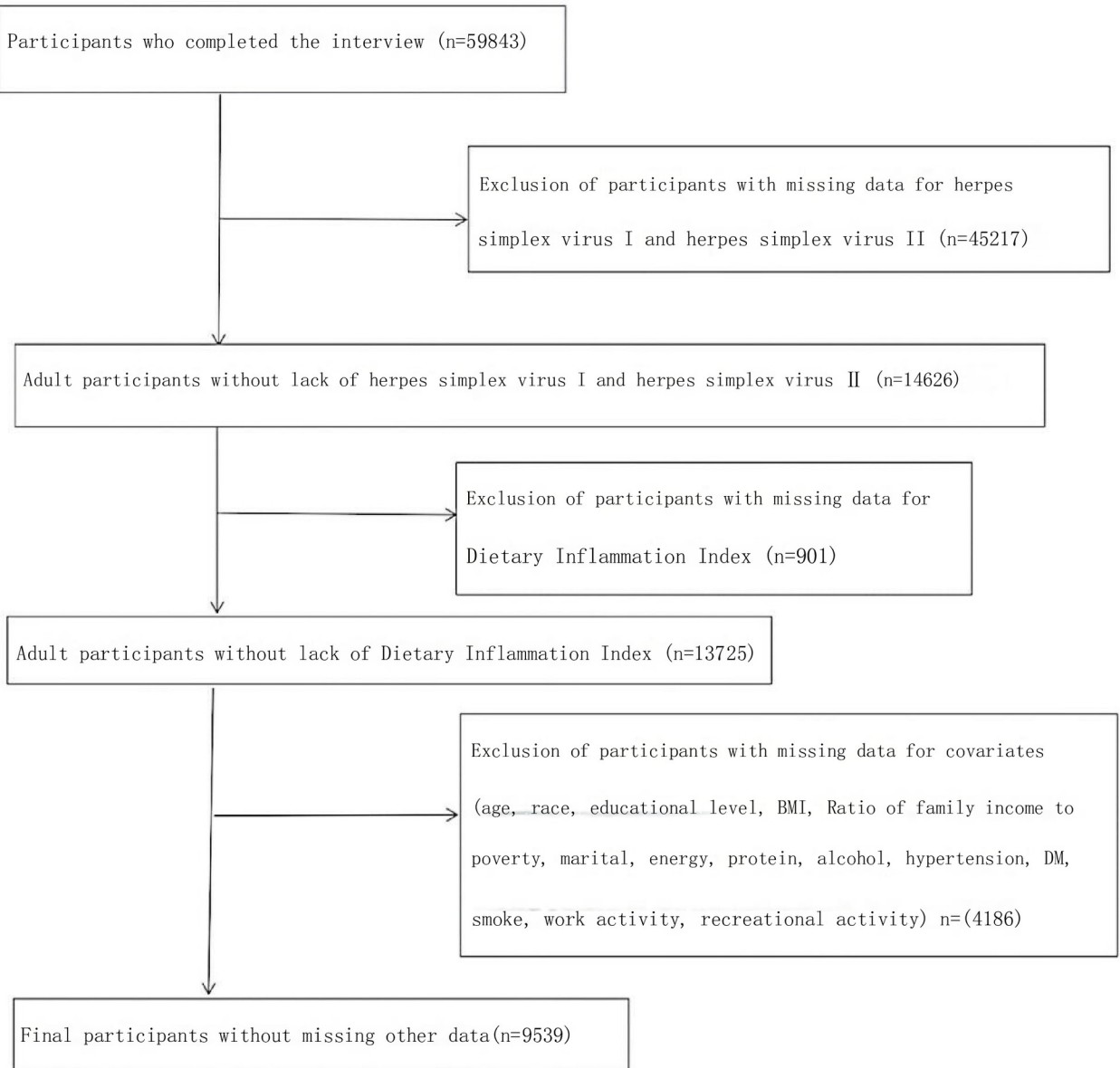

**Fig 1. Flow chart for participant inclusion.** Flow chart of participant enrollment for the analysis from the National and Nutrition Examination Survey 2007–2016 dataset.

## Measurement of herpes simplex virus

A mobile screening center collected blood samples by venipuncture, which were processed, stored, and shipped to Emory University for testing. Participants provided informed consent for the collection and processing of blood samples. Although extensive antigenic cross reactivity exists between herpes caused by HE-I and HE-II, two virus-specific glycoproteins, both gG-1 for HE-I and gG-2 for HE-II, have been identified to differentiate the two herpes viruses. To purify these glycoproteins, monoclonal antibodies and affinity chromatography were used, providing antigens for serological evaluation of the two herpesvirus strains. Antibodies that react to these antigens can be detected using solid-phase enzyme immunoassays. In the center of a nitrocellulose disk, purified glycoproteins gG-1 and gG-2 are adsorbed. Nonspecific

proteins were prevented from adhering to the remainder of the disk surface by coating it with bovine serum albumin. Specific antibodies allowed by incubation of the test serum, the discs, were bound to the immobilized antigen. As soon as nonreactive antibodies were thoroughly washed away, antibodies bound to goat anti-human IgG conjugated with peroxidase and an enzyme substrate (H2O2 with 4-chloro-1-naphthol) were detected. A positive reaction was indicated when a blue dot appeared in the center of the disc. Serum reacting to an immune spot filled with gG-1 indicated the presence of HSV-1 infection in the tested person, and serum reacting to an immune spot filled with gG-2 indicated the presence of HSV-2 infection in the tested person.

## Covariates

Several factors may affect our results. In our study, we selected age, education level, race, body mass index (BMI), household income to poverty ratio, marriage, energy, protein, hypertension, diabetes mellitus, smoking, work activity, and recreational activity as relevant covariates.

Participant age was obtained through a demographic questionnaire according to certain criteria, and respondents answered with a range of values, with age >85 coded as 85 years. Race/ethnicity was categorized as non-Hispanic black, non-Hispanic white, other Hispanic, Mexican American, and other race. The level of education is categorised as lower than high school, high school and higher than high school. BMI measurements are listed in the examination data. Participants' height and weight were determined according to the National Institutes of Health III anthropometric standards by adopting the correct technique, obtained by dividing weight in kilograms by height in meters (low BMI, high BMI, underweight and healthy weight ≤24.9, overweight and obese >25). The household income to poverty ratio (PIR) was collected on a demographic questionnaire.

To determine financial assistance eligibility for some federal assistance programs, NHANES uses the Department of Health and Human Services Poverty Guidelines (e.g., food vouchers, national school lunch programs). The variable was recoded into the following categorical values: high PIR (government assistance is available to participants who fall below 30% of the poverty line, which is roughly $25,000), medium PIR (with an annual income between $25k and $75k, they are considered middle class), and low PIR (considered high income with annual household incomes >$75,000). Marital status, energy, and protein, were collected from demographically relevant questionnaires, and NHANES collected the marital status of participants over the age of 14 for the study. Energy intake and protein intake were calculated by surveying participants' diets over a 24-hour period.

The Food and Nutrition Database for Dietary Studies 2.0 (FNDDS 2.0) of the USDA may be consulted. The report includes comprehensive information on coded foods and portion sizes reported by participants, as well as nutrient values for the calculation of nutrient intake. Participants' hypertension status was available in the form of a questionnaire. A questionnaire was administered to participants asking if they had ever been told they have high blood pressure (also known as hypertension) by a doctor or other health professional; if they self-reported antihypertensive medication use; or if they had high biometric values (systolic blood pressure ≥140 mm Hg and/or diastolic blood pressure ≥90 mm Hg). These three questions allowed participants to be determined to have hypertension. It is also possible to obtain three consecutive blood pressure readings at an ambulatory screening center; if blood pressure measurements are interrupted or incomplete, additional determinations may have been measured. In our study, we used the average of the readings to define hypertension. We classify the presence of diabetes into four categories: having diabetes, not having diabetes, impaired glucose tolerance (IGT) and impaired Fasting Glucose (IFG). The smoking history was also obtained

through relevant questionnaires or laboratory tests at an ambulatory screening center. As part of the Smoking Recent Tobacco Use Questionnaire, smoking status was assessed. By answering "yes" or "no" to the question "Tobacco/nicotine use in the past 5 days," participants were assessed for their smoking history. Work and recreational activities were used to obtain information from the questionnaire about the significance of participants' work and recreational activities on their health. Participants were asked questions about the different types of physical activity they engage in during a typical week to determine their work and recreational activities.

## Statistical analysis

This study employed R (http://www.R-project.org, The R Foundation) as the statistical software package and EmpowerRCH (free statistical software version 4.0) for all analyses. During data analysis, we determined the relationship between HE and DII and performed smoothed curve fitting, one-way analysis of variance, multiple regression equations, stratified analyses, and interaction tests to fully analyze the data. Numbers and percentages (%) are used to represent categorical variables, whereas mean + standard deviation or median (interquartile spacing) are used to represent continuous variables.

In this study, age, education level, race, body mass index (BMI), household income to poverty ratio, marriage, energy, protein, hypertension, diabetes, smoking, work activities, and recreational activities were stratified, and the p value results were approximately consistent across the above strata, indicating stability in the group and demonstrating that the correlation between DII and HE was significant.

The correlation between the dietary inflammation index and herpes simplex virus was represented by weighted multivariate logistic regression, where we adjusted the variables and divided them into crude and adjusted models depending on the number of adjusting variables added. The crude model does not adjust for any variables; Adjust 1 adjusts for age, education level, race, and BMI; Adjust 2 adjusts for age, education level, ethnicity, and BMI in Adjust 1; and Adjust 2 adds a family income poverty index and a family income poverty index and a family income poverty index to Adjust 1. Adjust 2 adds household income poverty index and marital status to Adjust 1; Adjust 3 adds hypertension, diabetes, smoking status, work activity, and leisure activity to Adjust 2; and Adjust 4 adds energy intake, and protein intake to Adjust 3. Herpes simplex virus types were divided into HE I and HE II. Smoothed curve fitting was used to visualize the association between the dietary inflammation index and herpes simplex virus, which showed that a DII of approximately 2 overlapped with HE1 at 0.60 and HE2 at 0.20, indicating an association; stratified analysis was the sensitivity analysis. Table 2 shows the association of DII with herpes simplex virus I. Table 3 shows the association of DII with herpes simplex virus II, Table 4 synthesizes the association of DII with HE, and Table 5 is the stratified analysis between each of the above adjusted variables. Univariate analysis is the evaluation of the covariates of association with outcome, and Table 2 shows the results of univariate analyses of education level, age, race, body mass index (BMI), household income to poverty ratio, marriage, hypertension, diabetes, smoking, work activity, energy, protein, and recreational activity. Relationships between HE were captured by multiple linear regression procedures. Interactions were examined by likelihood ratio tests between subgroups. To test for categorical variables and normal distributions, chi-square and t tests were used, and Kruskal–Wallis (skewed distribution) tests were used to assess continuous and categorical variables, respectively. In descriptive analysis, continuous variables are expressed as the mean and standard deviation or median and interquartile range (IQR), and categorical variables are expressed as weighted percentages (%). In addition, we calculated 95% confidence intervals (CL) with $p < 0.05$ as the level of statistical significance.

## Result

### Baseline characteristics of participants

This study used participant data from NHANES 2007–2016, including 9539 participants participating in the analysis in Fig 1. Herpes simplex virus type I was identified in 5867 participants, and herpes simplex virus type II was identified in 1948 participants. Table 1 demonstrates the baseline characteristics of the participants. Participants with herpes simplex virus type I were more likely to be older, female, Mexican, American, or other Hispanic, have a

**Table 1. The baseline characteristics of participants.**

| Characteristic | Herpes simplex virusI | | | Herpes simplex virus II | | |
|---|---|---|---|---|---|---|
| | Negative | Positive | P-value | Negative | Positive | P-value |
| N | 3673 | 5867 | | 7592 | 1948 | |
| DII | 1.23 (1.12, 1.34) | 1.49 (1.39, 1.59) | <0.0001 | 1.30 (1.20, 1.39) | 1.78 (1.63, 1.93) | <0.0001 |
| Age (year), | 32.90 (32.36, 33.43) | 35.99 (35.63, 36.35) | <0.0001 | 33.88 (33.45, 34.32) | 38.33 (37.74, 38.92) | <0.0001 |
| Sex | | | <0.0001 | | | <0.0001 |
| male | 46.30 (44.06, 48.56) | 53.65 (52.21, 55.09) | | 47.07 (45.74, 48.41) | 67.19 (64.37, 69.88) | |
| female | 53.70 (51.44, 55.94) | 46.35 (44.91, 47.79) | | 52.93 (51.59, 54.26) | 32.81 (30.12, 35.63) | |
| Race | | | <0.0001 | | | <0.0001 |
| Mexican American | 5.40 (4.33, 6.70) | 16.04 (13.38, 19.12) | | 11.82 (9.79, 14.21) | 8.65 (6.58, 11.27) | |
| Other Hispanic | 4.10 (3.11, 5.39) | 8.68 (7.01, 10.70) | | 6.06 (4.88, 7.49) | 9.58 (7.21, 12.63) | |
| Non-Hispanic White | 72.94 (69.86, 75.81) | 53.63 (49.44, 57.76) | | 65.25 (61.78, 68.56) | 46.88 (42.04, 51.79) | |
| Non-Hispanic Black | 9.71 (7.99, 11.75) | 12.77 (10.92, 14.88) | | 7.64 (6.43, 9.06) | 30.54 (26.57, 34.81) | |
| Other Race—Including Multi-Racial | 7.86 (6.67, 9.25) | 8.89 (7.65, 10.32) | | 9.24 (8.03, 10.60) | 4.35 (3.25, 5.81) | |
| Education level | | | <0.0001 | | | <0.0001 |
| <High school | 8.20 (6.86, 9.78) | 19.25 (17.34, 21.31) | | 13.01 (11.54, 14.64) | 21.01 (18.22, 24.10) | |
| High school | 17.38 (15.38, 19.58) | 23.26 (21.35, 25.29) | | 19.52 (17.72, 21.44) | 26.36 (23.35, 29.61) | |
| >High school | 74.42 (71.59, 77.06) | 57.49 (54.63, 60.30) | | 67.47 (64.73, 70.10) | 52.63 (48.69, 56.53) | |
| Ratio of family income to poverty | | | <0.0001 | | | <0.0001 |
| < = 1 | 14.09 (12.07, 16.38) | 21.50 (19.33, 23.84) | | 16.67 (14.78, 18.74) | 25.97 (22.48, 29.80) | |
| 1–3 | 32.79 (29.83, 35.89) | 39.03 (36.88, 41.22) | | 35.38 (33.20, 37.61) | 40.68 (36.67, 44.83) | |
| ≥3 | 53.13 (49.41, 56.81) | 39.47 (36.77, 42.25) | | 47.96 (44.93, 51.00) | 33.34 (28.84, 38.17) | |
| marital | | | 0.0040 | | | <0.0001 |
| Married | 56.97 (54.02, 59.87) | 56.25 (54.00, 58.48) | | 58.98 (56.97, 60.97) | 44.32 (40.56, 48.15) | |
| Widowed | 2.03 (1.41, 2.91) | 2.23 (1.71, 2.90) | | 1.98 (1.57, 2.48) | 2.98 (1.95, 4.54) | |
| Divorced | 8.35 (6.95, 10.00) | 9.81 (8.49, 11.32) | | 7.92 (6.82, 9.18) | 15.46 (12.99, 18.31) | |
| Separated | 2.01 (1.52, 2.66) | 3.02 (2.39, 3.81) | | 2.33 (1.89, 2.87) | 3.80 (2.78, 5.17) | |
| Never married | 20.95 (18.15, 24.06) | 17.00 (14.72, 19.56) | | 18.59 (16.36, 21.06) | 19.63 (16.66, 22.98) | |
| Living with partner | 9.68 (8.19, 11.42) | 11.68 (10.28, 13.24) | | 10.20 (9.05, 11.48) | 13.81 (11.35, 16.70) | |
| BMI (kg/m^2) | 28.14 (27.80, 28.47) | 29.08 (28.78, 29.37) | <0.0001 | 28.38 (28.10, 28.66) | 30.06 (29.52, 30.60) | <0.0001 |
| Energy (kcal) | 4528.55 (4449.87, 4607.23) | 4290.57 (4223.89, 4357.24) | <0.0001 | 4437.65 (4370.44, 4504.86) | 4187.77 (4076.10, 4299.44) | 0.0006 |
| Protein (g) | 178.47 (174.95, 181.99) | 169.05 (166.08, 172.02) | 0.0001 | 175.69 (172.93, 178.45) | 160.79 (156.71, 164.86) | <0.0001 |
| Hypertension | | | <0.0001 | | | <0.0001 |
| No | 83.65 (81.96, 85.22) | 78.43 (76.70, 80.07) | | 81.88 (80.56, 83.13) | 75.06 (72.25, 77.68) | |
| yes | 16.35 (14.78, 18.04) | 21.57 (19.93, 23.30) | | 18.12 (16.87, 19.44) | 24.94 (22.32, 27.75) | |
| Diabetes mellitus (DM) | | | <0.0001 | | | 0.0001 |
| No | 91.09 (89.59, 92.39) | 86.27 (84.83, 87.60) | | 88.97 (87.63, 90.18) | 85.59 (83.25, 87.65) | |

(*Continued*)

**Table 1.** (Continued)

| Characteristic | Herpes simplex virus I | | | Herpes simplex virus II | | |
|---|---|---|---|---|---|---|
| | Negative | Positive | P-value | Negative | Positive | P-value |
| DM | 4.12 (3.33, 5.09) | 7.17 (6.27, 8.19) | | 5.22 (4.57, 5.95) | 8.83 (7.30, 10.66) | |
| Impaired glucose tolerance (IGT) | 2.27 (1.57, 3.26) | 3.39 (2.76, 4.17) | | 2.98 (2.35, 3.78) | 2.44 (1.64, 3.61) | |
| Impaired Fasting Glucose (IFG) | 2.52 (1.86, 3.42) | 3.16 (2.53, 3.95) | | 2.83 (2.25, 3.55) | 3.14 (2.32, 4.23) | |
| Smoke | | | <0.0001 | | | <0.0001 |
| never | 65.76 (62.88, 68.52) | 57.17 (54.79, 59.52) | | 63.85 (61.44, 66.18) | 46.50 (43.08, 49.96) | |
| now | 19.34 (17.42, 21.42) | 25.36 (23.37, 27.45) | | 20.13 (18.46, 21.90) | 35.64 (32.15, 39.28) | |
| former | 14.90 (13.13, 16.87) | 17.47 (15.99, 19.06) | | 16.03 (14.63, 17.52) | 17.86 (15.23, 20.84) | |
| Work activity | | | 0.0104 | | | 0.7377 |
| no | 51.49 (48.82, 54.15) | 52.07 (49.93, 54.21) | | 51.45 (49.16, 53.72) | 53.67 (50.32, 56.99) | |
| both | 19.00 (16.61, 21.64) | 20.40 (18.88, 22.02) | | 19.87 (18.26, 21.58) | 19.29 (16.99, 21.82) | |
| moderate | 25.92 (23.88, 28.08) | 22.39 (20.77, 24.11) | | 24.20 (22.87, 25.58) | 22.79 (19.86, 26.02) | |
| vigorous | 3.59 (2.82, 4.57) | 5.13 (4.46, 5.89) | | 4.48 (3.87, 5.19) | 4.24 (2.96, 6.04) | |
| Recreational activity | | | <0.0001 | | | <0.0001 |
| no | 31.01 (28.73, 33.39) | 43.69 (41.16, 46.25) | | 36.31 (34.08, 38.59) | 46.86 (43.17, 50.57) | |
| both | 28.98 (26.55, 31.54) | 21.49 (19.37, 23.77) | | 26.22 (24.14, 28.41) | 17.72 (15.02, 20.79) | |
| moderate | 26.41 (24.35, 28.59) | 24.60 (22.77, 26.53) | | 25.07 (23.47, 26.74) | 27.13 (24.40, 30.04) | |
| vigorous | 13.60 (11.95, 15.43) | 10.22 (8.97, 11.62) | | 12.40 (11.28, 13.61) | 8.29 (6.73, 10.19) | |

Mean ± SD or median (IQR): P values were calculated by one-way ANOVA (normal distribution) and Kruskal–Wallis H (skewed distribution) test % for categorical variables. P values were calculated by the chi-square test.

higher BMI, more likely to smoke, have a lower household income to poverty ratio, have less education, consume less energy and protein and have less leisure activity. Interestingly, participants who were positive for herpes simplex virus type I had higher DII scores (1.23 (1.121.34) vs 1.49 (1.39–1.59), p<0.001). For herpes simplex virus type II, positive participants were possibly older, male, non-Hispanic black or other Hispanic, less educated, had a lower household income and poverty index, had a higher BMI, consumed less energy and protein, and had less leisure activity. Participants who were herpes simplex virus type II positive also had higher DII scores (1.30 (1.20,1.39) vs 1.78 (1.63,1.93), p<0.001). In addition, a higher percentage of positive participants had hypertension and diabetes, whether herpes simplex virus type I or type II.

## Association between DII and herpes simplex virus

The association between the DII and herpes simplex virus is shown in Tables 2–4. The DII was included in multiple logistic regression as an exposure factor in the form of tertiles. For herpes simplex virus type I, a positive association of DII with viral positivity was observed in the crude model. Unfortunately, however, the association became nonsignificant after adjusting for confounders. For herpes simplex virus type II, a positive association of DII with viral positivity was observed in all models. In the fully adjusted model, participants with high DII scores had 30% higher odds of suffering from herpes simplex virus II than those with low DII scores (OR = 1.30, 95% CI1.02, 1.65, p = 0.037).

## Stratified analysis

As shown in Table 5, variables found to be statistically significant in the univariate analyses were included in the stratified analyses, and confounders were adjusted according to the

**Table 2. Association between DII and herpes simplex virus I.**

| Exposure | Low | | Middle | | High | |
|---|---|---|---|---|---|---|
| | OR (95%CI) | P-value | OR (95%CI) | P-value | OR (95%CI) | P-value |
| Crude model | Ref. | | 1.25 (1.07, 1.21) | 0.006 | 1.34 (1.15, 1.55) | <0.001 |
| Adjust 1 | Ref. | | 1.19 (1.01, 1.41) | 0.047 | 1.27 (1.08, 1.50) | 0.005 |
| Adjust 2 | Ref. | | 1.17 (0.99, 1.39) | 0.078 | 1.22 (1.03, 1.44) | 0.021 |
| Adjust 3 | Ref. | | 1.14 (0.96, 1.37) | 0.137 | 1.18 (1.00, 1.40) | 0.054 |
| Adjust 4 | Ref. | | 1.04 (0.86, 1.27) | 0.663 | 1.00 (0.81, 1.23) | 0.968 |

Non adjusted model adjust for: None

Model I adjusted for age, educational level, race, and BMI.

Model II adjusted for Model I + Ratio of family income to poverty, marital

Model III adjusted for Model II + hypertension, DM, smoking, work activity, and recreational activity

Model IV adjusted for Model III +Energy, and Protein

**Table 3. Association between DII and herpes simplex virus II.**

| Exposure | Low | | Middle | | High | |
|---|---|---|---|---|---|---|
| | OR (95%CI) | P-value | OR (95%CI) | P-value | OR (95%CI) | P-value |
| Crude model | Ref. | | 1.61 (1.26, 2.05) | <0.001 | 1.90 (1.54, 2.34) | <0.001 |
| Adjust 1 | Ref. | | 1.41 (1.12, 1.78) | 0.004 | 1.57 (1.26, 1.95) | 0.001 |
| Adjust 2 | Ref. | | 1.35 (1.07, 1.72) | 0.002 | 1.41 (1.13, 1.75) | 0.003 |
| Adjust 3 | Ref. | | 1.33 (1.06, 1.67) | 0.017 | 1.35 (1.10, 1.66) | 0.007 |
| Adjust 4 | Ref. | | 1.22 (0.96, 1.55) | 0.110 | 1.15 (0.89, 1.48) | 0.294 |

Non adjusted model adjust for: None

Model I adjusted for: Age, Educational level, Race, BMI

Model II adjusted for Model I + Ratio of family income to poverty, marital

Model III adjusted for Model II + hypertension, DM, smoking, work activity, and recreational activity

Model IV adjusted for Model III +Energy and Protein

**Table 4. Association between DII and Simultaneous infection with herpes simplex virus I and II.**

| Exposure | Low | | Middle | | High | |
|---|---|---|---|---|---|---|
| | OR (95%CI) | P-value | OR (95%CI) | P-value | OR (95%CI) | P-value |
| Crude model | Ref. | | 1.61 (1.26, 2.05) | <0.001 | 1.90 (1.54, 2.34) | <0.001 |
| Adjust 1 | Ref. | | 1.41 (1.12, 1.78) | 0.004 | 1.57 (1.26, 1.95) | 0.001 |
| Adjust 2 | Ref. | | 1.35 (1.07, 1.72) | 0.002 | 1.41 (1.13, 1.75) | 0.003 |
| Adjust 3 | Ref. | | 1.33 (1.06, 1.67) | 0.017 | 1.35 (1.10, 1.66) | 0.007 |
| Adjust 4 | Ref. | | 1.22 (0.96, 1.55) | 0.110 | 1.15 (0.89, 1.48) | 0.294 |

Non adjusted model adjust for: None

Model I adjusted for: Age, Educational level, Race, BMI

Model II adjusted for Model I + Ratio of family income to poverty, marital

Model III adjusted for Model II + hypertension, DM, smoking, work activity, and recreational activity

Model IV adjusted for Model III +Energy, and Protein

**Table 5. Stratified analysis of the association between dietary inflammation index and herpes simplex virus.**

| X = DII | herpes simplex virus I | | | herpes simplex virus II | |
|---|---|---|---|---|---|
| | N | OR (95%CI) | P-value | OR (95%CI) | P-value |
| Age group | | | | | |
| 20–29 | 2979 | 0.97 (0.92, 1.02) | 0.1750 | 1.05 (0.96, 1.15) | 0.2586 |
| 30–39 | 3201 | 1.04 (0.99, 1.10) | 0.0867 | 1.05 (0.98, 1.12) | 0.1561 |
| 40–49 | 3359 | 0.99 (0.94, 1.04) | 0.6463 | 1.02 (0.96, 1.08) | 0.5402 |
| Sex | | | | | |
| male | 4994 | 1.02 (0.97, 1.06) | 0.4727 | 1.05 (1.00, 1.10) | 0.0614 |
| female | 4545 | 0.98 (0.94, 1.02) | 0.4344 | 1.04 (0.98, 1.10) | 0.2341 |
| Race | | | | | |
| Mexican American | 1587 | 0.96 (0.87, 1.05) | 0.3413 | 1.04 (0.93, 1.15) | 0.4917 |
| Other Hispanic | 960 | 0.91 (0.82, 1.02) | 0.0991 | 1.01 (0.91, 1.13) | 0.8286 |
| Non-Hispanic White | 3972 | 1.04 (1.00, 1.09) | 0.0702 | 1.02 (0.96, 1.09) | 0.5386 |
| Non-Hispanic Black | 1836 | 1.03 (0.97, 1.10) | 0.3495 | 1.06 (0.99, 1.13) | 0.1047 |
| Other Race—Including Multi-Racial | 1184 | 0.90 (0.84, 0.98) | 0.0116 | 1.21 (1.04, 1.40) | 0.0134 |
| Education level | | | | | |
| <High school | 1828 | 0.94 (0.86, 1.03) | 0.1614 | 0.97 (0.89, 1.06) | 0.4813 |
| High school | 2060 | 0.99 (0.92, 1.06) | 0.7958 | 1.03 (0.95, 1.12) | 0.4803 |
| >High school | 5651 | 1.02 (0.98, 1.05) | 0.3586 | 1.09 (1.03, 1.14) | 0.0015 |
| Ratio of family income to poverty | | | | | |
| < = 1 | 2344 | 1.02 (0.95, 1.09) | 0.5867 | 1.03 (0.96, 1.11) | 0.3982 |
| 1–3 | 3841 | 0.98 (0.93, 1.03) | 0.4125 | 1.08 (1.01, 1.14) | 0.0166 |
| ≥3 | 3354 | 1.01 (0.96, 1.06) | 0.6980 | 1.01 (0.95, 1.08) | 0.7537 |
| marital | | | | | |
| Married | 5241 | 1.00 (0.96, 1.04) | 0.9562 | 1.01 (0.96, 1.07) | 0.7064 |
| Widowed | 220 | 1.31 (1.05, 1.64) | 0.0180 | 0.85 (0.66, 1.09) | 0.2002 |
| Divorced | 823 | 1.00 (0.91, 1.11) | 0.9268 | 1.07 (0.95, 1.20) | 0.2473 |
| Separated | 322 | 1.07 (0.89, 1.28) | 0.4675 | 1.10 (0.89, 1.34) | 0.3784 |
| Never married | 1812 | 0.99 (0.93, 1.06) | 0.8072 | 1.05 (0.97, 1.15) | 0.2201 |
| Living with partner | 1121 | 0.93 (0.85, 1.02) | 0.1218 | 1.19 (1.06, 1.34) | 0.0025 |
| BMI group | | | | | |
| 14.1–25.09 | 3139 | 0.99 (0.94, 1.04) | 0.5988 | 1.07 (0.99, 1.15) | 0.0784 |
| 25.1–30.69 | 3202 | 0.97 (0.92, 1.02) | 0.2238 | 1.02 (0.96, 1.09) | 0.5434 |
| 30.7–73.43 | 3198 | 1.04 (0.99, 1.10) | 0.1204 | 1.05 (0.99, 1.12) | 0.1058 |
| ENERGY_KCAL group | | | | | |
| 18–3474 | 3179 | 1.02 (0.97, 1.08) | 0.4320 | 1.00 (0.94, 1.08) | 0.9102 |
| 3475–4815 | 3180 | 1.00 (0.96, 1.05) | 0.8922 | 1.08 (1.01, 1.15) | 0.0247 |
| 4816–20050 | 3180 | 1.00 (0.95, 1.05) | 0.9681 | 1.03 (0.97, 1.10) | 0.3084 |
| PROTEIN_G group | | | | | |
| 0.01–132.6 | 3180 | 1.02 (0.96, 1.08) | 0.5006 | 1.00 (0.93, 1.07) | 0.9448 |
| 132.61–188.65 | 3179 | 1.02 (0.97, 1.07) | 0.3975 | 1.09 (1.03, 1.17) | 0.0058 |
| 188.66–1014.99 | 3180 | 0.96 (0.92, 1.01) | 0.1293 | 1.04 (0.98, 1.11) | 0.1775 |
| Hypertension | | | | | |
| No | 7497 | 1.00 (0.96, 1.03) | 0.8366 | 1.06 (1.02, 1.11) | 0.0057 |
| yes | 2042 | 1.02 (0.96, 1.09) | 0.5611 | 0.98 (0.91, 1.06) | 0.5828 |
| DM | | | | | |
| No | 8210 | 0.99 (0.96, 1.02) | 0.5552 | 1.04 (1.00, 1.08) | 0.0665 |
| DM | 709 | 1.01 (0.90, 1.14) | 0.8146 | 1.05 (0.93, 1.20) | 0.4154 |

*(Continued)*

**Table 5.** (Continued)

| X = DII | herpes simplex virus I | | | herpes simplex virus II | |
|---|---|---|---|---|---|
| | N | OR (95%CI) | P-value | OR (95%CI) | P-value |
| IGT | 317 | 1.08 (0.89, 1.31) | 0.4147 | 1.17 (0.90, 1.52) | 0.2373 |
| IFG | 303 | 1.21 (0.99, 1.48) | 0.0593 | 1.05 (0.82, 1.34) | 0.6884 |
| Smoke | | | | | |
| never | 5883 | 1.00 (0.96, 1.03) | 0.8645 | 1.04 (0.99, 1.10) | 0.1038 |
| now | 2262 | 1.02 (0.96, 1.09) | 0.5636 | 1.06 (0.99, 1.14) | 0.1137 |
| former | 1394 | 0.97 (0.90, 1.04) | 0.3735 | 1.03 (0.93, 1.13) | 0.5985 |
| Work activity | | | | | |
| no | 5147 | 1.00 (0.97, 1.04) | 0.8456 | 1.05 (1.00, 1.10) | 0.0722 |
| both | 1791 | 1.04 (0.97, 1.11) | 0.2791 | 1.04 (0.95, 1.13) | 0.3757 |
| moderate | 2157 | 0.97 (0.92, 1.04) | 0.3894 | 1.03 (0.95, 1.12) | 0.4723 |
| vigorous | 444 | 0.95 (0.82, 1.11) | 0.5448 | 1.08 (0.89, 1.32) | 0.4213 |
| Recreational activity | | | | | |
| no | 4148 | 1.00 (0.96, 1.05) | 0.9089 | 1.00 (0.95, 1.06) | 0.8680 |
| both | 2012 | 1.01 (0.95, 1.07) | 0.7640 | 1.03 (0.94, 1.12) | 0.5429 |
| moderate | 2287 | 1.02 (0.96, 1.08) | 0.5658 | 1.10 (1.02, 1.19) | 0.0148 |
| vigorous | 1092 | 0.94 (0.86, 1.02) | 0.1440 | 1.13 (0.99, 1.29) | 0.0729 |

The adjusted model in the stratification analysis was constructed based on Adjust 4, adjusted for age, educational level, race, BMI, ratio of family income to poverty, marital, hypertension, DM, smoke, work activity, recreational activity, energy, and protein

adjustment model 4. Stratified analyses demonstrated the association of each DII unit increase with herpes simplex virus type I and type II in different populations. For herpes simplex virus type I, similar positive associations were observed in the vast majority of subgroups. For herpes simplex virus type II, similar positive associations were observed in all subgroups. Overall, the results of the stratified analyses suggest that the positive association of DII with herpes simplex virus type II is stable in the population.

## Discussion

In this cross-sectional study, we found a positive association between DII and the risk of HSV. In addition, it was found that DII and vitamin C have an interactive effect in increasing NAFLD.

As we know, this is a cross-sectional study that studied the interaction of DII on the association of vitamin C intake and HSV for the first time. Diet may act on the development of HCC through its potential inflammation. HSV infection causes an inflammatory reaction in the body [11]. In prior research, higher DII/E-DII scores have been associated with several inflammation-related diseases [12]. Previous studies have examined the influence of specific dietary components on HSV. Studies have shown that confusing enough vitamin E in one's diet is conducive to immune protection against herpes simplex encephalitis and other viral infections [13, 14]. Supplementary vitamin D reduces HSV-1 virus levels and mRNA performance in infected HSV-1 cells [15]. A clinical study revealed that for an unknown number of individuals, ingesting 23 mg of zinc (sulfate) and 250 mg of vitamin C twice a day for six weeks could minimize the length and severity of the attack [16].

In addition, different dietary flavonoids have different effects on HSV. Catechins inhibited the infectivity of the HSV-1 virus but had no inhibitory effect on replication and no significant effect on other viruses. Hesperetin did not affect the infectivity of HSV but reduced the

intracellular replication of HSV. Quercetin led to a concentration-dependent decrease in the infectivity of HSV. In addition, the effect of quercetin on the intracellular replication of HSV is influenced by the mode of infection [17]. Myricetin inhibited HSV infection and its replication by downregulating the EGFR/PI3K/Akt signaling pathway [18]. All of these nutrients are known to affect the DII [6]. Various mechanisms may explain these findings, but inflammation may be a crucial mediator.

In addition, diet may affect HSV infection in other ways. CD8+ T cells are an important factor in treating HSV-1 infection in the central nervous system and play a vital role in responding to HSV-1 [19, 20]. Dietary vitamin E levels regulate T regulatory and dendritic cells in the periphery [13]. In the life cycle of viruses, HSV entry is the key stage of infection and cell-to-cell transmission. The current treatment methods have difficulty eliminating the virus after the initial stage of this infection [21]. For example, myricetin may interfere with virus adsorption and membrane fusion by directly interacting with the virus gD protein to block HSV infection [18]. New therapies aimed at preventing entry will be more helpful in preventing the virus from spreading to new hosts [22]. If acyclovir and other anti-herpetic drugs are used for a long time, they will cause drug-resistant strains and several side effects [23]. Dietary manipulation is an effective way to limit the severity of viral immune-inflammatory lesions [24].

Vitamin C, which has important antioxidant, immune regulatory, antiviral, antithrombotic, and anti-inflammatory traits, is an essential nutrient [25–27]. Studies have proven that vitamin C inactivates all kinds of viruses in vitro, including HSV, influenza A, and poliovirus 1 [28, 29]. The intake of vitamin C strengthens the immune system and can protect the body against diseases [30]. A radiation experiment showed that 2-O-glycoside of ascorbic acid has remarkable antiviral properties against HSV-1 and inhibits the rupture and recombination reactions to the alpha-hydroxyl effect [31]. However, under physiological stress conditions such as infection, trauma, and surgery, the level of vitamin C in human plasma drops rapidly [32].

Our research has some limitations. First, due to the cross-sectional design, we cannot prove the causal or directional relationship between DII and HSV. Even after multiple adjustments, other unmeasured variables may still affect the results. Second, dietary data, RIP, marital status, work activity, recreational activity, and smoking status based on interviews or self-report questionnaires from NHANES are inevitable. Moreover, only US residents were included in the study, so it needs to be considered when extrapolating to other populations. Therefore, it is necessary to conduct multicenter, carefully designed controlled trials to validate our findings.

## Conclusion

This study demonstrated a positive association between DII and herpes simplex virus type II in US adults, suggesting that a proinflammatory diet may be an independent risk factor for herpes simplex virus type II. However, the underlying mechanisms by which food drives herpes simplex virus type II infections require further study.

## Author Contributions

**Conceptualization:** Jing Luo, Xu-Guang Guo.

**Data curation:** Jing Luo, En-Hui Liu, Hao-Kai Chen.

**Methodology:** Jing Luo.

**Project administration:** Jing Luo, En-Hui Liu.

**Supervision:** Jing Luo.

**Visualization:** Xu-Guang Guo.

**Writing – original draft:** Jing Luo, En-Hui Liu, Hao-Kai Chen, Xiang-Ping He, Tong Chen, Yu-Qi Hu.

**Writing – review & editing:** Jing Luo, En-Hui Liu, Hao-Kai Chen, Xiang-Ping He, Tong Chen, Yu-Qi Hu.

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
