## [Decision Letter · Decision Letter 0]

23 Apr 2024

PONE-D-24-13567Association between dietary inflammation index and herpes simplex virus I and II: a cross-sectional studyPLOS ONE

Dear Dr. Guo,

Thank you for submitting your manuscript to PLOS ONE. After careful consideration, we feel that it has merit but does not fully meet PLOS ONE’s publication criteria as it currently stands. Therefore, we invite you to submit a revised version of the manuscript that addresses the points raised during the review process.

Your manuscript has been reviewed by reviewers, and they have provided their comments on your study. Please consider revising your manuscript by addressing the comments from both reviewers and kindly replying to them.

We look forward to receiving your revised manuscript.

Kind regards,

Engin Berber, D.V.M., Ph.D.

Academic Editor

PLOS ONE

Journal Requirements:

3. Please include your tables as part of your main manuscript and remove the individual files. Please note that supplementary tables (should remain/ be uploaded) as separate "supporting information" files

Reviewers' comments:

Reviewer's Responses to Questions

**Comments to the Author**

1. Is the manuscript technically sound, and do the data support the conclusions?

Reviewer #1: Yes

Reviewer #2: Partly

2. Has the statistical analysis been performed appropriately and rigorously? 

Reviewer #1: I Don't Know

Reviewer #2: No

3. Have the authors made all data underlying the findings in their manuscript fully available?

Reviewer #1: Yes

Reviewer #2: Yes

4. Is the manuscript presented in an intelligible fashion and written in standard English?

Reviewer #1: Yes

Reviewer #2: No

5. Review Comments to the Author

Reviewer #1: Dear Authors

I read your manuscript carefully; your work will be accepted with minor revision. Please apply the following comments.

Article entitled “Association between dietary inflammation index and herpes simplex virus I and II: a cross-sectional study” has been written in a good way; It is an interesting topic. Authors only address following minor revisions:

Abstract:

-Your keywords should write based on MeSH term.

Introduction

In the introduction, you should first discuss the role of dietary intake (especially natural supplements and complementary medicine) in the virus infection diseases (COVID 19, Herpes and…), then discuss the probably association between dietary inflammatory index with HSV prevalence. You can also use the following references in your article for strengthen the introduction and discussion sections. These references are helpful.

1) Shan, T., Huang, Y., Zhao, Z., Li, F., Wang, Y., Ye, C., ... & Ren, Z. (2023). Ketogenic diet restrains herpes simplex encephalitis via gut microbes. Microbes and Infection, 25(3), 105061.

2) Evaluation of the effect of kefir supplementation on inflammatory markers and clinical and hematological indices in COVID-19 patients; a randomized double-blind clinical trial (Advances in Integrative Medicine).

3) Effect of a diet based on Iranian traditional medicine on inflammatory markers and clinical outcomes in COVID-19 patients: A double-blind, randomized, controlled (European Journal of Integrative Medicine).

4) Current Nutritional Support in Critical Ill Covid-19 Patients: A Brief Review (DOI:10.26420/austincritcarej.2021.1034).

5) Effects of Melatonin and Propolis Supplementation on Inflammation, Oxidative Stress, and Clinical Outcomes in Patients with Primary Pneumosepsis: A Randomized Controlled Clinical Trial (Complementary medicine research).

6) Firouzi et al. The effect of Vitamin C and Zn supplementation on the immune system and clinical outcomes in COVID-19 patients.

7) Does propolis have any effect on rheumatoid arthritis? A review study (Food Science & Nutrition).

-Given that similar studies have been done before, please mention the logic and novelty of your work.

-Please mention the novelty of your study.

Methods:

- Please more explain about your study design.

- Did the authors calculate study sample prior the study?

- Do you have information on the household socio-economic status? It has been demonstrated that "generally" adherence to a healthier pattern is associated with higher educational and socio-economic level of the family.

- I couldn't see the food intake chart (micronutrients), is it measured?

Results:

Well presented.

Discussion:

Authors should improve discussion section by implementing recent evidence. You can also use the above references.

- I found some grammatical errors in your work. Please revise in again by a native person.

Best Regard

Reviewer #2: The author conducted extensive research to explore the association between the Dietary Inflammation Index and herpes simplex virus I and II based on the NHANES datasets. However, there are many questions here that may need further improvement.

1) In the Abstract-Introduction, "to fill....and to provide...." changed to "We aimed to fill....and to provide.... "

2) Why choose data from these five cycles in the NHANES (2007-2008, 2009-2010, 2011-2012, 2013-2014,

95 and 2015-2016)?

3) line 160: "Education level was classified as high school graduate, college, less than high school

and above". This sentence is a bit confusing and does not align with the following table. In thel table, education level was classified into >high school, high school, and <high school.="">4)line 188: How is diabetes detected? According to the following table, diabetes status were classified as "no", "DM", "IGT", and "IFG". Please provide detailed definitions for these indicators.

5) In Table 2, the header row should include OR (95% CI).

6) Importantly, NHANES employed a complex, multistage, probability sampling design. Using NHANES weights can make the study results more representative, reflecting the overall situation of the US population rather than just individuals in the sample. Therefore, the author should conduct weighted logistic regression analysis when performing this study. As for selecting and calculating weights, it is advisable for the author to consult relevant literature.</high>

6. PLOS authors have the option to publish the peer review history of their article (what does this mean?). If published, this will include your full peer review and any attached files.

Reviewer #1: No

Reviewer #2: No

---

## [Author Response · Author response to Decision Letter 0]

25 Aug 2024

Response to Reviewers

Many thanks to the reviewers for their suggestions, and here are the revisions for each point.

To reviewer 1：

1. we found dietary inflammation index is not in MeSH, referring to other published articles on dietary inflammation index, we still put it in keywords, about which we can discuss further.

2. Thank you for your suggestion, it has been revised according to your suggestion.

3. Regarding the innovativeness of our article, we pointed out in the introduction of the manuscript that "No studies have yet examined the correlation between DII and herpes simplex virus infections among U.S. adults. S. adults. Therefore, exploring the correlation between DII and mono infections will provide new insights into the prevention and management of mono infections in the future. infections in the future."

4. the research design of our article is described in Statistical Analysis of Methods (line 248)

5. The study sample has been added (line 129)

6. grammatical errors in the discussion section have been corrected. 

To reviewer 2：

1. the presentation of Abstract-Introduction has been revised

2. because DII-related data are only available for these five cycles in NHANES web.

3. the statement about education level has been revised (line 197).

4. specific diabetes testing indicators are mentioned in the official website of NHANES, do they need to be written in the body of the article? Diabetes related explanation has been modified.

5. Graphs and charts have been modified

---

## [Decision Letter · Decision Letter 1]

18 Oct 2024

PONE-D-24-13567R1Association between dietary inflammation index and herpes simplex virus I and II: a cross-sectional studyPLOS ONE

Dear Dr. Guo,

Thank you for submitting your manuscript to PLOS ONE. After careful consideration, we feel that it has merit but does not fully meet PLOS ONE’s publication criteria as it currently stands. Therefore, we invite you to submit a revised version of the manuscript that addresses the points raised during the review process.

Please revise your draft accordingly and address the reviewers' comments.

We look forward to receiving your revised manuscript.

Kind regards,

Engin Berber, D.V.M., Ph.D.

Academic Editor

PLOS ONE

Reviewers' comments:

Reviewer's Responses to Questions

**Comments to the Author**

1. If the authors have adequately addressed your comments raised in a previous round of review and you feel that this manuscript is now acceptable for publication, you may indicate that here to bypass the “Comments to the Author” section, enter your conflict of interest statement in the “Confidential to Editor” section, and submit your "Accept" recommendation.

Reviewer #2: (No Response)

Reviewer #3: All comments have been addressed

2. Is the manuscript technically sound, and do the data support the conclusions?

Reviewer #2: (No Response)

Reviewer #3: Yes

3. Has the statistical analysis been performed appropriately and rigorously? 

Reviewer #2: (No Response)

Reviewer #3: Yes

4. Have the authors made all data underlying the findings in their manuscript fully available?

Reviewer #2: (No Response)

Reviewer #3: Yes

5. Is the manuscript presented in an intelligible fashion and written in standard English?

Reviewer #2: (No Response)

Reviewer #3: Yes

6. Review Comments to the Author

Reviewer #2: 1. In the definition of diabetes as a covariate, you are not asked to classify diabetes into four types, but rather to distinguish between the presence or absence of diabetes based on multiple indicators from NHANES. A fasting blood glucose level above 7.0 mmol/L can be defined as diabetes, but if it is below 7.0 mmol/L, does that mean the participants doesn't have diabetes? You also need to consider whether they are taking blood sugar-lowering medications or receiving other glucose-lowering treatments. Additionally, it’s important to look at HbA1c levels, such as whether it’s greater than 6.5%. All of these factors should be considered together to determine whether the individual has diabetes.These data can be obtained from the NHANES database.

2. Importantly, NHANES employed a complex, multistage, probability sampling design. Using NHANES weights can make the study results more representative, reflecting the overall situation of the US population rather than just individuals in the sample. Therefore, the author should conduct weighted logistic regression analysis when performing this study.

Reviewer #3: This study entitled Association between dietary inflammation index and herpes simplex virus I and II: a cross-sectional study is interesting. However, I have some comments.

1. The result section in abstract is so general without any emphasizing on number or for example OR or CI . it should be corrected.

2. I think it would have been better to talk about the disease first in the introduction section and then talk about the inflammatory index of the diet in the following paragraphs.

3. In the introduction section, it is better to mention in one sentence the association between the inflammatory diet and the risk of acute and chronic diseases (you can read these studies and use as references in this section 10.1016/j.dsx.2015.09.015 and 10.1002/ptr.7081

4. Research Gap and Novelty: The claim that no studies have examined DII and HSV among U.S. adults is important but should be emphasized more. Moreover, explain why the NHANES dataset is uniquely suited for this study.

5. You should explain more the exclusion criteria.

6. When you calculate DII, did you adjust the effects of energy for every 45 components?

7. The smoothing curve fitting should be explained more

8. Its better to mention the key findings in each model with specific p-values and odds ratios to guide readers through the statistical significance and strength of associations more effectively.

7. PLOS authors have the option to publish the peer review history of their article (what does this mean?). If published, this will include your full peer review and any attached files.

Reviewer #2: No

Reviewer #3: No

---

## [Author Response · Author response to Decision Letter 1]

7 Nov 2024

Response to Reviewers

Reviewer #2 

1 Thank you very much for your advice, the stages of diabetes can be categorized into four, having diabetes, not having diabetes, impaired glucose tolerance (IGT) and impaired Fasting Glucose (IFG), by subdividing the diabetes stages to adjust for variables, as well as further stratified analyses at follow-up. In addition, there is more missing data for HbA1c, with a missing count of 8889 for 2007-2008, and the small sample size may not be conducive to analyzing the relationship between DII and herpes simplex virus.

2 Thank you for your suggestion, very professional, the data for our logistic regression analysis was weighted but missed count, now it has been added.

Reviewer #3

1. Thank you very much for your valuable suggestions, have added OR and CI values to the abstract section.

2. Thank you for your valuable suggestion, the introduction section has been modified.

3. Thank you for your suggestions, relevant references have been cited

4. Thank you for your valuable suggestions on this study. Regarding study gaps and novelty, this has been further clarified. In particular, the lack of studies on the relationship between DII and HSV infection among US adults has been emphasized. In addition, regarding the selection of the NHANES dataset, we have also provided more details in the manuscript.

5.We have added an explanation of the exclusion criteria in the Methods section, in addition to an explanation of all variables in the covariates section.

6. yes, we have adjusted for 45 components in the calculation of DII, of which 28 are food related.

7. The smoothed curve fit for herpes simplex virus and DII may be difficult to clearly see the correlation due to the small sample size, so we did not add it in the article.

8. Thank you for your suggestion, we have added specific p-values and other data.

---

## [Decision Letter · Decision Letter 2]

18 Dec 2024

Association between dietary inflammation index and herpes simplex virus I and II: a cross-sectional study

PONE-D-24-13567R2

Dear Dr. Guo,

We’re pleased to inform you that your manuscript has been judged scientifically suitable for publication and will be formally accepted for publication once it meets all outstanding technical requirements.

Kind regards,

Engin Berber, D.V.M., Ph.D.

Academic Editor

PLOS ONE

Additional Editor Comments (optional):

Reviewers' comments:

Reviewer's Responses to Questions

**Comments to the Author**

1. If the authors have adequately addressed your comments raised in a previous round of review and you feel that this manuscript is now acceptable for publication, you may indicate that here to bypass the “Comments to the Author” section, enter your conflict of interest statement in the “Confidential to Editor” section, and submit your "Accept" recommendation.

Reviewer #2: All comments have been addressed

Reviewer #3: All comments have been addressed

Reviewer #4: All comments have been addressed

2. Is the manuscript technically sound, and do the data support the conclusions?

Reviewer #2: Yes

Reviewer #3: Yes

Reviewer #4: Yes

3. Has the statistical analysis been performed appropriately and rigorously? 

Reviewer #2: Yes

Reviewer #3: Yes

Reviewer #4: Yes

4. Have the authors made all data underlying the findings in their manuscript fully available?

Reviewer #2: Yes

Reviewer #3: Yes

Reviewer #4: Yes

5. Is the manuscript presented in an intelligible fashion and written in standard English?

Reviewer #2: Yes

Reviewer #3: Yes

Reviewer #4: Yes

6. Review Comments to the Author

Reviewer #2: (No Response)

Reviewer #3: I think authors could make all of the comments in the new version of manuscript and it has good quality for acceptance.

Reviewer #4: The manuscript is well written, and it provides valuable data to fill the gap. I have no comments to raise concern.

7. PLOS authors have the option to publish the peer review history of their article (what does this mean?). If published, this will include your full peer review and any attached files.

Reviewer #2: No

Reviewer #3: No

Reviewer #4: No

---

## [Editor Report · Acceptance letter]

14 Jan 2025

PONE-D-24-13567R2 

PLOS ONE

Dear Dr. Guo, 

I'm pleased to inform you that your manuscript has been deemed suitable for publication in PLOS ONE. Congratulations! Your manuscript is now being handed over to our production team.

Kind regards, 

on behalf of

Dr. Engin Berber 

Academic Editor

PLOS ONE